# Differences in Internet Use and eHealth Needs of Adolescent and Young Adult Versus Older Cancer Patients; Results from the PROFILES Registry

**DOI:** 10.3390/cancers13246308

**Published:** 2021-12-16

**Authors:** Carla Vlooswijk, Olga Husson, Emiel J. Krahmer, Rhodé Bijlsma, Suzanne E. J. Kaal, Sophia H. E. Sleeman, Lonneke V. van de Poll-Franse, Winette T. A. van der Graaf, Nadine Bol, Mies C. van Eenbergen

**Affiliations:** 1Research and Development, Netherlands Comprehensive Cancer Organisation, 3511 DT Utrecht, The Netherlands; C.Vlooswijk@iknl.nl (C.V.); L.vandePoll@iknl.nl (L.V.v.d.P.-F.); M.vanEenbergen@iknl.nl (M.C.v.E.); 2Department of Psychosocial Research and Epidemiology, Netherlands Cancer Institute, 1066 CX Amsterdam, The Netherlands; 3Department of Medical Oncology, Netherlands Cancer Institute—Antoni van Leeuwenhoek, 1066 CX Amsterdam, The Netherlands; w.vd.graaf@nki.nl; 4Department of Surgical Oncology, Erasmus MC Cancer Institute, Erasmus University Medical Center, 3015 GD Rotterdam, The Netherlands; 5Division of Clinical Studies, Institute of Cancer Research, London SM2 5NG, UK; 6Tilburg Center for Cognition and Communication (TiCC), Department of Communication and Cognition, Tilburg University, 5037 AB Tilburg, The Netherlands; E.J.Krahmer@tilburguniversity.edu (E.J.K.); Nadine.Bol@tilburguniversity.edu (N.B.); 7Department of Medical Oncology, University Medical Center, 3584 CX Utrecht, The Netherlands; R.M.Bijlsma@umcutrecht.nl; 8Department of Medical Oncology, Radboud University Medical Center, 6525 GA Nijmegen, The Netherlands; Suzanne.Kaal@radboudumc.nl; 9Dutch AYA ‘Young and Cancer’ Care Network Utrecht, 3511 DT Utrecht, The Netherlands; sophia@ayazorgnetwerk.nl; 10Center of Research on Psychology in Somatic Diseases, Department of Medical and Clinical Psychology, Tilburg University, 5037 AB Tilburg, The Netherlands; 11Department of Medical Oncology, Erasmus Cancer Institute, Erasmus Medical Center, 3015 GD Rotterdam, The Netherlands

**Keywords:** AYAs, internet use, eHealth, cancer patients

## Abstract

**Simple Summary:**

The internet has become an important health information source for patients with cancer. AYAs (adolescents and young adults; 18–39 years at time of diagnosis) can be considered as digital natives; they work with the internet and related technologies in their daily lives. It is likely that AYAs are more used to using the internet, while older cancer patients might prefer former ways of information provision to obtain health-related information. The question arises whether internet use and eHealth needs of AYA cancer patients are comparable to those of older ones. By conducting a cross-sectional survey, we evaluated differences in cancer-related internet patterns between AYAs and older cancer patients (40+ years at time of diagnosis). A better understanding of differences between generations will help inform healthcare providers on how to guide cancer patients of different ages regarding cancer-related internet use.

**Abstract:**

Background: Our aim was to evaluate differences in cancer-related internet patterns between AYAs (adolescents and young adults; 18–39 years at time of diagnosis) and older adult cancer patients (40+ years). Methods: Cross-sectional surveys were distributed among AYA and older adult cancer patients regarding cancer-related internet use and eHealth needs. Results: 299 AYAs (mean age 31.8 years) and 270 older adults (mean age 55.8 years) participated. AYAs searched significantly more often on the internet on a daily basis just before diagnosis (45% vs. 37%), right after diagnosis (71% vs. 62%) and during treatment (65% vs. 59%) compared to older adults. During follow up, there was a trend that AYAs searched less often on the internet compared to older adults (15% vs. 17%). AYAs searched more often on topics, such as alternative or complementary therapies, treatment guidelines, fertility, end of life, sexuality and intimacy, lifestyle and insurance. AYAs felt significantly better informed (75%) after searching for cancer-related information compared to older adults (65%) and had significantly less unmet needs regarding access to their own medical information (22% vs. 47%). AYAs search more on the internet on a daily basis/several times per week in the diagnosis and treatment phase than older cancer patients. They search on different topics than older adults and seems to have less unmet eHealth needs.It is important that these are easy to find and reliable.

## 1. Introduction

Over the past decades, the internet has become an important source of health information for patients, and can be accessed instantaneously at any time [1]. At an increasing number of hospitals, patients can access their medical information in Electronic Health Records (EHR) and communicate online with healthcare professionals. In addition, these days, patients can read and post content in online health communities, interact via social media, and share information with other patients very easily.

In 2017, 85% of the patients with cancer in the Netherlands used the internet on a daily basis [2]. The majority (72%) of patients with cancer who searched online for information about cancer felt more informed about their disease after consulting the internet [3]. Patients who feel more informed may feel more equipped to take part in the medical decision-making process and may feel more confident in interacting with healthcare professionals [4]. A systematic review showed that self-monitoring of side effects of cancer treatment was associated with a positive effect on patient-centered outcomes [5]. In 2020, van Eenbergen et al. demonstrated that 72% of the patients in the Netherlands wish to access their EHR, and approximately 33% actually accessed their EHR.

However, not all patients are alike, and some may require different online information and resources compared to others. In particular, the question arises whether internet use and wishes of younger cancer patients are comparable to those of older ones.

In the Netherlands, approximately 4% (i.e., 4000 persons per year) of all new cancer diagnoses occur among adolescents and young adults (AYAs) between 18 and 39 years. AYAs face unique physical, emotional and social challenges due to their age and developmental stage of life [6]. Cancer challenges the abilities of AYAs’ to achieve milestones, which can lead to stagnation of their normal development. Typical cancer issues such as premature confrontation with mortality, being more dependent on others, changes in one’s physical appearance, disruptions in study, work, or social life because of the treatment, and potential loss of fertility are particularly distressing [7].

How and why patients use the internet depends on factors such as gender, socioeconomic status and age [2,8]. In this manuscript, the different usage of internet between ages will be studied. It is plausible that online information needs differ between AYAs (18–39 years at diagnosis) and older adults (40+ at diagnosis). AYAs can be considered a digital savvy generation, i.e., digital natives. AYAs work with the internet (social media, online communities) and related technologies in their daily lives. Internet use, and specifically health-related internet use is higher among AYAs compared to older adults [9]. It is likely that AYAs are more used to using the internet, while older cancer patients might prefer former ways of information provision to obtain health-related information. A better understanding of differences between generations will inform healthcare providers on how to best guide cancer patients of different ages with cancer-related internet use. Therefore, we evaluated differences in cancer-related internet use patterns between AYAs and older cancer patients. To investigate these differences, we examined the quantity, content, perceived impact of cancer-related internet use and eHealth needs.

## 2. Materials and Methods

We conducted two cross-sectional studies via PROFILES (‘Patient Reported Outcomes Following Initial treatment and Long term Evaluation of Survivorship’) [10]. The study which included AYAs (18–39 years old at time of diagnosis) was conducted in 2019 and the study which included older adults (40+ years old at time of diagnosis) was conducted in 2016 and 2017 (Appendix A).

### 2.1. AYAs (18–39 Years at Diagnosis)

AYA cancer patients were invited to complete an online questionnaire. Participants with unknown age at time of diagnosis (*n* = 10) or diagnosed before 2010 (*n* = 294) were excluded from analyses. Patients diagnosed before 2010 were excluded because the internet was less accessible and less commonly used, since internet access and use increased each year [9]. This study was approved by the Research Ethics Review Committee of Tilburg School of Humanities and Digital Sciences (internal code: REDC 2019.104). AYAs were invited through different methods of recruiting, and based on this recruiting method divided in two groups: population-based population (POP) and online cancer community group (OCC).

#### 2.1.1. AYAs—POP

AYAs who were already participating in the population-based SURVAYA study were invited to participate and complete an online questionnaire for this study. The SURVAYA study is a population-based study conducted in the academic hospitals and Antoni van Leeuwenhoek hospital in the Netherlands. AYAs who were diagnosed between 1999 and 2015 with cancer were identified through the Netherlands Cancer Registry (NCR).

#### 2.1.2. AYAs—OCC

AYAs were invited via the kanker.nl online community, newsletters of the Dutch Federation of Cancer Patient Organizations (NFK) and other patient platforms.

### 2.2. Older Adults (40+ Years at Diagnosis)

Cancer patients were invited to complete a questionnaire. Patients who were younger than 40 years at age at time of diagnosis (*n* = 17) or had missing data on age at time of diagnosis (*n* = 18) were excluded. Since prostate cancer patients were not present in the AYA sample, older cancer participants with prostate cancer (*n* = 97) were excluded. Cancer patients who were diagnosed before 2010 (*n* = 44) were excluded. A declaration of no objection was granted by the Medical Ethics Review Committee Midden Brabant, (NW2016-47). Older adults were invited through different methods of recruiting, and based on this recruiting method divided in two groups: population-based population (POP) and online cancer community group (OCC).

#### 2.2.1. Older Adults—POP

A population-based sample of breast and gynecological cancer and lymphoma patients was identified through the Netherlands Cancer Registry (NCR). Patients who were diagnosed between 2014 and 2016 in 3 hospitals in the Netherlands and between the ages of 18 and 70 years old at the time of diagnosis. They were invited via their physician to participate in the study and completed the questionnaire on paper.

#### 2.2.2. Older Adults—OCC

Older adults with different kinds of tumor types who were member of the Kanker.nl platform were recruited by e-mail to fill in the same online questionnaire.

Since no validated Dutch questionnaire on cancer internet use existed, we developed one in 2004 [3]. This questionnaire was based on a literature study and the four internet functions defined by Eysenbach, namely communication, content, community and e-commerce [11]. In 2017, some of the questions were updated due to developments of internet in the intermittent years, including increased access to eHealth and blended care [2]. Patients were asked about the frequency of cancer-related internet use, searched topics online, impact of cancer-related internet use and eHealth needs. The impact of cancer-related internet use was assessed by the questions: ‘Did you feel better informed about your illness after consulting the internet?’, ‘Do you think that consulting your doctor has increased visits to your doctor?’, ‘Have you discussed the information you found on the internet with your healthcare provider(s)?’ and ‘Do you have the impression that the information found has influenced the choice of your therapy/treatment?’. EHealth needs were assessed by the questions: ‘What online options did you have?’ and ‘What would you use regarding the following eHealth topics?: access medical information, access own test results, e-consult physicians, e-consult nurses, request prescriptions, request tests, request referral, make appointment, perform test-self-diagnoses, online peers contact, receiving reminders and suggest ideas.’

Demographic (i.e., date of birth and gender) and clinical data (i.e., cancer type, primary treatments received and date of diagnosis) were obtained from the NCR when available (2.1.1; AYAs POP Group, 2.1.2; AYAs OCC Group, 2.2.1; Older Adults POP Group) or self-reported (2.2.2). The time between the diagnosis and completion of the questionnaire was determined by the difference in patients’ date of birth and completion date of the questionnaire. Cancer type was classified according to the third International Classification of Diseases for Oncology [12] when available (2.1.1; AYAs POP Group, 2.1.2; AYAs OCC Group, 2.2.1; Older adults POP group) or self-reported (2.2.2; Older Adults—OCC). Primary treatments received were classified into surgery, chemotherapy, radiotherapy, hormone therapy, immunotherapy, stem cell transplantation, targeted therapy or other. Educational level (primary school/secondary school/college/university) and marital status were assessed by the questionnaire.

### 2.3. Statistical Analysis

Statistical analyses were conducted using SAS version 9.4 (SAS Institute, Cary, NC, USA, 1999) and two-sided *P*-values of <0.05 were considered statistically significant. All variables were described as percentages or means and standard deviations. Differences between AYAs and older adults were compared using chi-square analyses for categorical variables (or Fishers exact tests when sample sizes are small) and analyses of variance (ANOVAs) for continuous variables. Since the outcomes were different for the different groups based on method of recruiting (AYAs—POP, AYAs—OCC, older adult POP and older adults—OCC), a sensitivity analysis was performed (Appendix A).

## 3. Results

In total, 569 participants participated in the study including 299 AYAs (mean age: 31.8 years) and 270 older adults (mean age: 55.8 years). Table 1 shows the demographic and clinical characteristics of the respondents. At time of participation in the study, AYAs were significantly longer after diagnosis (6.1 years) compared to older adults (3.1 years) (<0.0001). Moreover, AYAs were significantly higher educated (<0.0001), less often had gynecological cancer (0.0012) and lung cancer (0.0105), and were primarily treated more often with chemotherapy (0.0350) and stem cell transplantation (0.0028) and less often with targeted therapy (<0.0001).

### 3.1. Cancer Related Internet Use

Figure 1 shows that AYAs searched significantly more often information about cancer on the internet on a daily basis/several times a week just before diagnosis (0.0463), right after diagnosis (0.0008) and during treatment (0.0144) than older adults. In contrast, during follow up, there was a trend that AYAs searched less often on the internet compared to older adults (0.0591). 

The POP groups of both AYAs and older adults searched the internet less often compared to the OCC groups. But in these POP groups, AYAs searched significantly more often than older adults on a daily basis/several times a week right after diagnosis (0.0057) and during treatment (0.0091). The OCC group of AYAs was more active on the internet just before diagnosis, right after diagnosis and during treatment compared to the OCC group of the older adults, but this was not significantly different. To the contrary, during follow-up, older adults searched more often on the internet than AYAs, but this was not significant different (Appendix A).

### 3.2. Content of Cancer-Related Internet Use

Older adults searched significantly more often for information on type of cancer (86% vs. 64% < 0.0001), financial problems (27% vs. 13% < 0.0001) and meeting possibilities with peers (43% vs. 24% < 0.0001) (Table 2). AYAs searched significantly more often for information on alternative or complementary therapies (32% vs. 22% 0.0077), treatment guidelines (65% vs. 56% 0.0210), fertility (47% vs. 2% < 0.0001), end of life (26% vs. 15% 0.0016), sexuality and intimacy (52% vs. 38% 0.0019), lifestyle (72% vs. 55% < 0.0001) and legal regulations (insurance) (44% vs. 28% < 0.0001).

### 3.3. Perceived Impact of Cancer-Related Internet Use

Figure 2 shows that AYAs felt significantly better informed (75%) after searching for cancer-related information on the internet compared to older adults (65%) (0.0112). Thereby, AYAs discussed the information more often with the physician (83% vs. 80% 0.4262), but this was not significant different. Respectively 2% and 1% (0.5116) of AYAs and older adults reported that their doctor visits increased through searching for cancer-related information on the internet. Moreover, no significant difference (0.1682) was observed between the AYAs (18%) and older adults (13%) regarding the influence of searching for cancer-related information on internet on their treatment choice.

The POP group of AYAs felt significantly better informed (70%) after searching for cancer-related information on the internet compared to the POP group of older adults (50%) (0.0006) (Appendix A). Thereby, they discussed the information more often with the physician (78% vs. 72% 0.1921). Around 2% of all the participants reported that their doctor visits increased through searching for cancer-related information on the internet. Moreover, no significant differences were observed between the groups regarding the influence of searching for cancer-related information on the internet on their treatment choice.

### 3.4. eHealth Needs

AYAs reported significantly less unmet needs regarding access to medical information (22% vs. 47% < 0.0001), their own test results (25% vs. 50% < 0.0001), and e-consult physicians (38% vs. 55% 0.0008), requesting tests (30% vs. 63% < 0.0001), requesting a referral (21% vs. 55% < 0.0001), making an appointment (38% vs. 51% 0.0083), performing test/self-diagnoses (13% vs. 49% < 0.0001), receiving reminders (22% vs. 45% < 0.0001) and suggesting ideas (33% and 60% < 0.0001) (Figure 3).

The OCC group of the older adults reported more unmet eHealth needs compared to the POP group of older adults and the POP and OCC group of AYAs, except for make an appointment (Appendix A). The POP group of the older adults reported significantly more unmet eHealth needs regarding access own test results, request tests, request referral, perform test/self-diagnosis and suggest ideas compared to the POP group of the AYAs.

## 4. Discussion

In this cross-sectional study, we evaluated the differences in cancer-related internet use patterns between AYAs and older cancer patients. We found that AYAs searched more on the internet on a daily basis/several times per week in the diagnosis and treatment phase of their disease. AYAs searched more often for alternative or complementary therapies, treatment guidelines, fertility, end of life, sexuality and intimacy, lifestyle and insurances, whereas the older adults searched more often for information about types of cancer, cancer genetics and heritability, financial problems and meeting possibilities with peers. Older adults seem to have more unmet eHealth needs ethan AYAs.

AYAs reported that they searched more often on cancer-related information than older adults. Dobransky and Hargittai suggest that those who are more skilled in using the internet, are more likely to use it for health information seeking [13]. This was in line with the findings of a study in Canada where older adults less often used the internet, and were also less confident in their ability to evaluate online information on cancer-care decision making than younger cancer patients [14]. This could be a possible explanation for the differences in cancer-related internet use in our study. 

In our study, AYA cancer patients searched more often for online topics, such as fertility, end of life, lifestyle and insurances than older adults. This is in line with other studies and also logically explains why AYAs searched more often on age-specific topics like fertility [15]. The differences in searching on lifestyle and legal relations and insurances between AYAs and older adults is a more remarkable outcome. Other studies have also reported that AYA cancer patients use the internet to search for healthy lifestyle behaviors, but recommend adjusting internet resources to better fit the needs of AYAs [16]. Namely, AYAs indicated that they had difficulties finding information tailored to their experience as a AYA, desired age-specific recommendations, and were concerned about the trustworthiness of internet sites. In addition, our study found that older adults searched more often for clinical trials online compared to AYAs, which is in line with the underrepresentation of AYAs participating in clinical trials [17].

The impact of searching for cancer-related information on the internet was significantly different for AYAs and for older adults. Where 65% of older adults felt more informed after searching on the internet, only 75% of the AYAs did. It is important that healthcare professionals should be aware of the kind of information AYAs and older adults searched for on the internet and guide them to reliable and accurate high-quality internet sources [18]. It is essential that patients are informed about the potential risks of internet use. Misleading or misinterpreted health information on the internet may compromise health behaviors and health outcomes and could potentially negatively impact patient–healthcare professional relationships.

This study included POP and OCC patients. In line with our study, previous research has shown that OCC patients are not fully representative of the total population of cancer patients; therefore, it is plausible to believe that OCC patients are not fully comparable to general population (POP)-based samples [19]. Therefore, additional sensitivity analyses were performed (Appendix A). Our study results show that OCC patients inherently searched more often daily/several times per week on the internet and were higher educated. It seems reasonable to assume that they have more health literacy skills than the general population. These factors are important to keep in mind when conducting and interpreting future research. Moreover, our AYA sample included a larger proportion of highly educated cancer patients compared to the older adult sample. The lower educated AYA sample includes a greater number of patients who have low (health) literacy skills [20]. It is important to understand and assess health related information while making health-related decisions. Therefore, low educated AYAs may need more guidance related to cancer-related internet use.

There are several limitations of the current work. First, as the questionnaire was not available on paper for AYAs and the OCC group of older adults, they were solely invited to participate in the study via an online questionnaire. In contrast, the POP group of the older adults could complete the questionnaire on paper. This might have led to selection bias, in which more active internet users were included in the AYA sample and the OCC-group of older adults. Second, AYAs were diagnosed with cancer longer ago than the older adults patients in our study. The expansion of internet usage over the last several decades might have led to an underestimation of internet use in AYAs and an overestimation of the unmet needs of AYAs. Moreover, recall bias might have occurred in AYAs due to the longer interval between cancer diagnosis and completing the questionnaire. Third, the distribution of cancer types in the sample of AYAs and older adults differed significantly. However, we do not think this is a major limitation as previous research has demonstrated, information needs and internet use do not differ between cancer types [21,22].

## 5. Conclusions

AYAs searched more on the internet on a daily basis/several times per week in the diagnosis and treatment phase of their disease than older adults and have less unmet eHealth needs. Furthermore, AYAs are more interested in topics online like alternative or complementary therapies, treatment guidelines, fertility, end of life, sexuality and intimacy, lifestyle and insurances, whereas the older adults searched more often for information about types of cancer, cancer genetics and heritability, financial problems and meeting possibilities with peers. It is important that these topics are easy to find and are reliable. OCC patients reported different internet usage than POP patients, which indicates that OCC patients are not representative of the total cancer population in terms of cancer-related internet use. Moreover, as our sample has on average a high educational level, it should be noted that lower educated AYAs may need more guidance related to cancer-related internet use than higher educated AYAs, for example more visual information. Accordingly, lower educated AYAs with cancer were underrepresented in our study and future studies should examine their online needs in more detail.

## Figures and Tables

**Figure 1 cancers-13-06308-f001:**
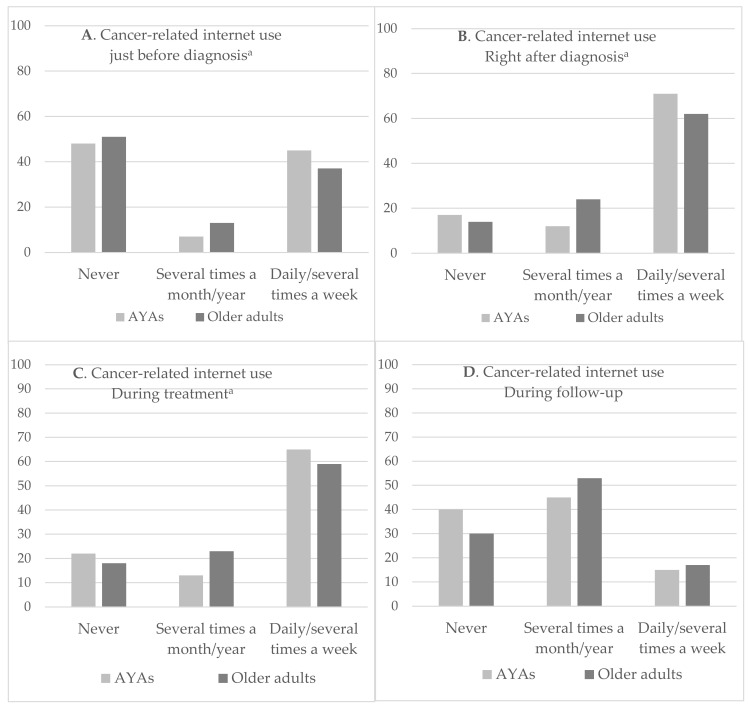
(**A**–**D**) Search frequency for information about cancer on the internet during different phases of disease according to AYAs and older adults. ^a^ significant difference between AYAs and older adults.

**Figure 2 cancers-13-06308-f002:**
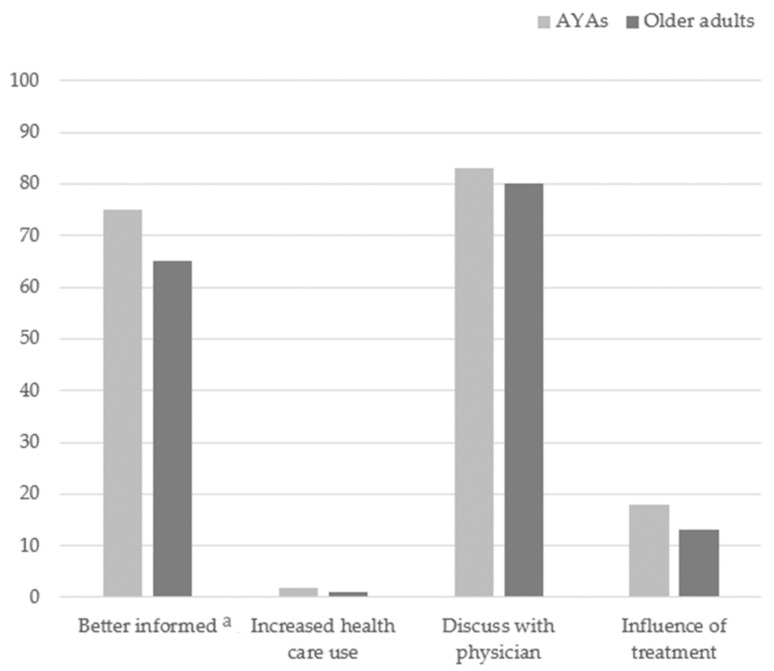
Impact of searching cancer-related information on the internet for AYAs and older adults. ^a^ significant difference between AYAs and older adults.

**Figure 3 cancers-13-06308-f003:**
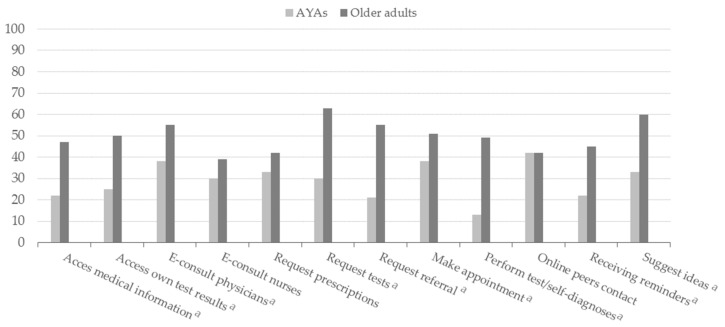
Unmet eHealth needs by AYAs and older adults. ^a^ significant difference between AYAs and older adults.

**Table 1 cancers-13-06308-t001:** Characteristics of the AYAs (*n* = 299) and older adults (*n* = 270).

Variation		Adolescents and Young Adults (18–39 Years)		Older Adults (40 + Years)		
		*N*	%	*N*	%	*p*-Value
Gender	Female	234	78	221	82	0.2852
	Male	65	22	49	18	
Age (at diagnosis), (mean (sd))		31.8 (5.7)		55.8 (9.2)		<0.0001
	18–24 years	45	15	0	0	
	25–39 years	254	85	0	0	
	40–64 years	0	0	112	88	
	65 + years	0	0	16	13	
Years since diagnosis (mean ± SD)		6.1 (2.9)		3.1 (1.4)		<0.0001
	0–2 year(s)	51	17	89	33	
	>2–5 years	54	18	160	59	
	>5 years	194	65	21	8	
Education level	Primary school	0	0	3	1	<0.0001
	Secondary school	108	36	169	63	
	College/university	190	64	96	36	
Marital status (at time of questionnaire)	Partner	233	78	220	82	0.2533
Type of cancer	Brain cancer	16	5	0	0	NA
	Breast cancer	105	35	111	41	0.1412
	Bone cancer	4	1	0	0	NA
	Gastrointestinal cancer	13	4	21	8	0.0848
	Gynacological cancer	43	14	68	25	0.0012
	Head and neck cancer	2	1	0	0	NA
	Leukemia	15	5	0	0	NA
	Lung cancer	3	1	12	4	0.0105
	Lymphoma	46	15	44	16	0.7660
	Sarcoma	8	3	0	0	NA
	Skin cancer	9	3	7	3	0.7636
	Testicular cancer	20	7	0	0	NA
	Thyroid cancer	12	4	0	0	NA
	Urological cancer	4	1	0	0	NA
	Other ^1^	7	2	7	3	0.8467
Treatment modality	Surgery	215	72	189	70	0.6168
	Chemotherapy	211	71	168	62	0.0350
	Radiotherapy	164	55	131	49	0.1312
	Hormone therapy	56	19	68	25	0.0625
	Immunotherapy	33	11	27	10	0.6876
	Stem cell transplantation	15	5	2	1	0.0028
	Targeted therapy	8	3	34	13	<0.0001
	Other ^2^	7	2	4	1	0.4571

^1^ Neuroendocrine tumor, mesothelioma, trophoblast tumor, multiple myeloma, esthesioneuroblastoma and thymus cancer. ^2^ Radioactive iodine therapy and no therapy or active surveillance. NA = not applicable.

**Table 2 cancers-13-06308-t002:** Medical and psychosocial topics searched for on the internet during and after treatment by AYAs and older adults.

Variation	Adolescents and Young Adults (18–39 Years) *n* = 299		Older Adults (40 + of Age) *n* = 270		*p*-Value
**Medical topics**	N	%	N	%	
Type of cancer	184	64	221	86	<0.0001
Treatments	225	78	203	79	0.7391
Consequences of treatment in general	228	79	201	79	0.9361
Alternative or complementary therapies	92	32	53	22	0.0077
Finding a doctor	72	25	50	20	0.1828
Finding a hospital	92	32	67	27	0.1824
Trials and/or research	85	30	93	37	0.0588
Treatment guidelines	188	65	140	56	0.0210
Cancer genetics and heritability	173	60	134	53	0.1180
Fertility	136	47	6	2	<0.0001
End of life	74	26	36	15	0.0016
**Psychosocial topics**	N	%	N	%	
What can I do myself	195	68	150	61	0.0638
Sexuality and intimacy	147	52	95	38	0.0019
Lifestyle	205	72	138	55	<0.0001
Legal regulations (insurance)	124	44	68	28	<0.0001
Financial problems	38	13	67	27	<0.0001
Meeting possibilities peers	67	24	108	43	<0.0001

## Data Availability

The data presented in this study are available on request from the corresponding author. The data are not publicly available due to privacy issues.

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
