# Peer review of "Differences in Internet Use and eHealth Needs of Adolescent and Young Adult Versus Older Cancer Patients; Results from the PROFILES Registry"

_cancers, 2021, doi:10.3390/cancers13246308_

Round 1

Reviewer 1 Report

This manuscript describes a novel and interesting study of AYA cancer patients (18-39 years) internet use and preferences for health information and resources access through the internet, as compared to older cancer patients (40+). Overall, the paper is well written and provides insight into an important topic in the digital era.

  1. In the title, and throughout the paper, it is unclear what “wishes” refers to. Consider rewording for clarity
  2. Given the broad definition of AYAs globally, suggest defining the eligible age range as early as possible including the Simple Summary and Abstract
  3. The Simple Summary has an unconvincing rationale/significance – how will knowing the proportion of survivors using the internet, and the topics that they search, help HCPs to guide cancer patients on their cancer-related internet use? The same rationale is applied in the introduction which undermines the novelty and significance of the study.
  4. In the abstract results, suggest including a brief example or two of “unmet needs regarding e-health”. The following concluding remark seems obvious/redundant – consider rewording (perhaps rephrasing “age-specific”) or removing “They are specifically more interested in age-specific topics online than older adults.”
  5. Introduction: “In the Netherlands, approximately 4% (i.e. 4,000 persons) of all new cancer diagnoses 76 occur among adolescents and young adults (AYAs) between 18 and 39 years” Is this 4,000 per year?
  6. Methods: Were there any additional inclusion/exclusion criteria for participating survivors (e.g. all diagnoses included? Any age at the time of survey completion? Any specific centers?). The method of identifying and inviting participants to complete the survey is particularly unclear.
  7. Older adults were recruited as early as 2016 vs AYAs who were recruited in 2019. Given the rapidly growing proportion of people using the internet, could this account for some of the lower internet use in older survivors? Another potential limitation worth noting is that males were underrepresented
  8. “In contrast, during follow up, AYAs searched less often on the Internet compared to older adults 184 (0.0591).” This is an interesting finding, and worth noting in the abstract. This could account for some of the lower reported levels of engagement in healthcare use and survivorship care reported in AYAs
  9. More information is needed about the measures included in the survey and the response options (E.g. to measure frequency of use, information needs, eHealth wishes?). In some cases, it is not clear that something is being measured at all until it appears in the results (e.g. discussing information with physicians). Appreciate that some of the detail has been published elsewhere, however sufficient data should be provided in this manuscript too for completeness
  10. Figure 1 – suggest adding clearer titles to each of the figures 1A-D (e.g. “Cancer-related internet use just before diagnosis”
  11. Table 2 – what does “age-specific” refer to? It seems to only apply to the psychosocial topics? If some survivors weren’t asked about all topics, this should be specified in the methods.
  12. Appropriate chi squared (or other) test statistics and p-values should be presented throughout the results, and consistently
  13. Section 3.4 – this sentence appears to be missing a word and should be revised for clarity of expression.
  14. What does “suggest ideas” refer to?
  15. Page 10, lines 278 – 286 – this paragraph highlights that fewer adult cancer patients felt informed after searching the internet, yet the discussion here focuses on the needs of providing AYAs with reliable/accurate information, which may be even more important for adult cancer patients if their needs are less likely to be met with the information available
  16. Did the authors consider exploring factors associated with internet use, information needs and preferences of each of the groups to assess if similar factors in AYA and adult cancer patients influenced their use/needs?

Author Response

We are very pleased to get the opportunity to improve our manuscript and would like to thank the reviewer for the useful comments. Changes are highlighted in the manuscript and are described below.

Reviewer #1

This manuscript describes a novel and interesting study of AYA cancer patients (18-39 years) internet use and preferences for health information and resources access through the internet, as compared to older cancer patients (40+). Overall, the paper is well written and provides insight into an important topic in the digital era.
Response: Thank you for your kind words and thank you for taking the time to review this manuscript so thoroughly. We really appreciate this.

1.    In the title, and throughout the paper, it is unclear what “wishes” refers to. Consider rewording for clarity
Response: Thank you for this comment. Because wishes refer to the need of e-health (section 3.4) we changed ‘wishes’ to ‘e-health’ needs throughout the paper to make it more clear.

2.    Given the broad definition of AYAs globally, suggest defining the eligible age range as early as possible including the Simple Summary and Abstract
Response: Thank you for this suggestion. We added ’18-39 years at time of diagnosis’ in the simple summary and abstract as early as possible. 

3.    The Simple Summary has an unconvincing rationale/significance – how will knowing the proportion of survivors using the internet, and the topics that they search, help HCPs to guide cancer patients on their cancer-related internet use? The same rationale is applied in the introduction which undermines the novelty and significance of the study.
Response: We changed the rationale/significance in the simple summary: ‘The internet has become an important health information source for patients with cancer. AYAs  (adolescents and young adults; 18-39 years at time of diagnosis)) can be considered as digital natives; they work with the internet and related technologies in their daily lives. It is likely that AYAs are more used to using the internet, while older cancer patients might prefer former ways of information provision to obtain health-related information. The question arises whether internet use and e-health needs of AYA cancer patients are comparable to those of older ones. 
We have also changed the introduction: ‘AYAs can be considered a digital savvy generation, i.e. digital natives. AYAs work with the internet (social media, online communities) and related technologies in their daily lives. Internet use, and specifically health-related internet use is higher among AYAs compared to older adults [9]. It is likely that AYAs are more used to using the Internet, while older cancer patients might prefer former ways of information provision  to obtain health-related information.’

4.    In the abstract results, suggest including a brief example or two of “unmet needs regarding e-health”. The following concluding remark seems obvious/redundant – consider rewording (perhaps rephrasing “age-specific”) or removing “They are specifically more interested in age-specific topics online than older adults.”
Response: We  added a brief example of unmet needs regarding e-health in the abstract: ‘AYAs felt significantly better informed (75%) after searching for cancer-related information compared to older adults (65%) and have less unmet needs regarding access to their own medical information (22% vs 47%).’
Thereby, we have removed ‘age-specific’. 

5.    Introduction: “In the Netherlands, approximately 4% (i.e. 4,000 persons) of all new cancer diagnoses 76 occur among adolescents and young adults (AYAs) between 18 and 39 years” Is this 4,000 per year?
Response: This is indeed 4,000 per year. We changed this in the introduction.

6.    Methods: Were there any additional inclusion/exclusion criteria for participating survivors (e.g. all diagnoses included? Any age at the time of survey completion? Any specific centers?). The method of identifying and inviting participants to complete the survey is particularly unclear.
Response: The inclusion and exclusion criteria were different for AYAs and older adults. There are stated in the method under the sections 2.1. AYAs and 2.2. older adults. There were no additional inclusion/exclusion criteria for participating survivors. 

7.    Older adults were recruited as early as 2016 vs AYAs who were recruited in 2019. Given the rapidly growing proportion of people using the internet, could this account for some of the lower internet use in older survivors? Another potential limitation worth noting is that males were underrepresented. 
Response: We asked the participants to fill in the questionnaire about internet usage at time of cancer diagnosis. Therefore, we think that the difference in recruiting date of older adults and AYAs not account for some of the lower internet use in older survivors. Moreover, internet use was in 2016 already high and did not really changed between 2016 and 2019.   

8.    “In contrast, during follow up, AYAs searched less often on the Internet compared to older adults 184 (0.0591).” This is an interesting finding, and worth noting in the abstract. This could account for some of the lower reported levels of engagement in healthcare use and survivorship care reported in AYAs
Response: We added the following text in the results part of the abstract: ‘During follow up, there was a trend that AYAs search less often on the internet compared to older adults (15% vs 17%).’  

9.    More information is needed about the measures included in the survey and the response options (E.g. to measure frequency of use, information needs, eHealth wishes?). In some cases, it is not clear that something is being measured at all until it appears in the results (e.g. discussing information with physicians). Appreciate that some of the detail has been published elsewhere, however sufficient data should be provided in this manuscript too for completeness
Response: This is indeed not clearly stated in the method section. We have add the following information to the method section:: 
‘Patients were asked about the frequency of cancer-related internet use, searched online topics online, impact of cancer-related internet use and e-health needs. The impact of cancer-related internet use was assessed by the questions: Did you feel better informed about your illness after consulting the internet? Do you think that consulting your doctor has increased visits to your doctor? Have you discussed the information you found on the internet with your healthcare provider(s)? Do you have the impression that the information found has influenced the choice of your therapy / treatment? E-health needs were assessed by the questions: What online options did you have and what wishes/what would you use regarding the following e-health topics: access medical information, access own test results, e-consult physicians, e-consult nurses, request prescriptions, request tests, request referral, make appointment, perform test-self-diagnoses, online peers contact, receiving reminders and suggest ideas.’.

10.    Figure 1 – suggest adding clearer titles to each of the figures 1A-D (e.g. “Cancer-related internet use just before diagnosis”
Response: Thank you for this suggestion. We added ‘cancer-related internet use’ in the titles of figure 1. 

11.    Table 2 – what does “age-specific” refer to? It seems to only apply to the psychosocial topics? If some survivors weren’t asked about all topics, this should be specified in the methods.
Response: We removed ‘age-specific’. The same topics were asked for AYAs and older adults.

12.    Appropriate chi squared (or other) test statistics and p-values should be presented throughout the results, and consistently
Response: We added p-values in section 3.3. to make the presentation of the results consistent. The significant findings in the graphs and tables are indicated with an ‘a’.

13.    Section 3.4 – this sentence appears to be missing a word and should be revised for clarity of expression.
Response: We added the word ‘to’: AYAs reported significantly less  unmet needs regarding to access medical information (22% vs 47% <.0001), access own test results (25% vs 50% <.0001), e-consult physicians (38% vs 55% 0.0008), request tests (30% vs 63% <.0001), request referral (21% vs 55% <.0001), make appointment (38% vs 51% 0.0083), perform test/self-diagnoses (13% vs 49% <.0001), receiving reminders (22% vs 45% <.0001) and suggest ideas (33% and 60% <.0001) (Figure 3).

14.    What does “suggest ideas” refer to?
Response: ‘Suggest ideas’ in section 3.4. refer to the possibility for patients to propose ideas online, for example ideas to improve health care.   

15.    Page 10, lines 278 – 286 – this paragraph highlights that fewer adult cancer patients felt informed after searching the internet, yet the discussion here focuses on the needs of providing AYAs with reliable/accurate information, which may be even more important for adult cancer patients if their needs are less likely to be met with the information available
Response: We changed this paragraph: ‘Where 65% of the older adults felt more informed after searching on the internet, 75% of the AYAs did. It is important that healthcare professionals should be aware of the kind of information AYAs and older adults searched for on the internet and guide them to reliable and accurate high-quality internet sources [18].’

16.    Did the authors consider exploring factors associated with internet use, information needs and preferences of each of the groups to assess if similar factors in AYA and adult cancer patients influenced their use/needs?
Response: Exploring factors associated with internet use, information needs and preferences was not the aim of this study. Moreover, factors influencing their use/needs are inherently different between the groups (for example tumor types), which make it less interesting and reliable to compare these outcomes between the groups. 

Reviewer 2 Report

This is a well written paper evaluating internet use in cancer patients, comparing internet use in cancer patients of different age groups, AYA vs. older adults.   As expected, overall internet use was higher in AYA patients.

Questions/grammar edits:

The authors mention EHR use in the introduction, but that doesn't appear to be part of the data collected.  If available, that would be interesting information to compare between the 2 groups

Simple summary: Methods-rewrite, doesn't make sense as currently written

Introduction, first sentence: rewrite as "...and can be accessed instantaneously at any time."

Materials and Methods: Section 2.1.1,   change sentence to "....is a population based study conducted in the academic hospitals....." 

Materials and Methods: Section 2.1.2: Were the AYAs contact by email like the older adults were?

Results, first paragraph: change "at the time of participating in the study" to "at the time of participation in the study"

Results, table 1: Wasn't significantly significant, but there was a female preponderance, and idea why?

Results, table 1:  Why do you think the AYA patients were off therapy a significantly longer period of time?

Results, table 2: There was a trend toward more internet searches for clinical trials in older adults.  Given the under-representation of AYA on clinical trials, I find this interesting and may be worth mentioning in the discussion.

Results, section 3.4: Remove the word "often" from the first sentence.

Discussion: Paragraph 4, line 180-change the word "was" to "did"

Discussion: Paragraph 5, line 291-the sentence beginning with "From the results of the study....." doesn't read well as currently written.  Please re-write to improve flow of the paper.

Author Response

Title: Differences in internet use and wishes of adolescent and young adult versus older cancer patients; results from the PROFILES registry

We are very pleased to get the opportunity to improve our manuscript and would like to thank the reviewer for the useful comments. Changes are highlighted in the manuscript and are described below.

Reviewer #2

This is a well written paper evaluating internet use in cancer patients, comparing internet use in cancer patients of different age groups, AYA vs. older adults.   As expected, overall internet use was higher in AYA patients.
Response: Thank you very much for the kinds words and reviewing our paper.

Questions/grammar edits:
The authors mention EHR use in the introduction, but that doesn't appear to be part of the data collected.  If available, that would be interesting information to compare between the 2 groups. 
Response: Thank you for this comment. In figure 3 we presented the unmet online information needs regarding E-health by AYAs and older adults. The first two bars show ‘acces medical information’ and ‘access own test results’. The unmet needs on these two topics are significantly higher for older adults compared to AYAs. 

Simple summary: Methods-rewrite, doesn't make sense as currently written
Response: We added in the simple summary that the study is a cross sectional survey: ‘By conducting a cross-sectional survey, we evaluated differences in cancer-related internet patterns  between AYAs and older cancer patients (40+ years at time of diagnosis).’

Introduction, first sentence: rewrite as "...and can be accessed instantaneously at any time."
Response: Thank you for this comment. We re-wrote this in the first sentence of the introduction: ‘Over the past decades, the internet has become an important source of health information for patients, and can be accessed instantaneously at any time.’

Materials and Methods: Section 2.1.1,   change sentence to "....is a population based study conducted in the academic hospitals....." 
Response: Thanks you for this comment. We changed the sentence to; is a population-based study conducted in the academic hospitals.

Materials and Methods: Section 2.1.2: Were the AYAs contact by email like the older adults were?
Response: Both group of AYAs (POP and OCC) were invited to complete the questionnaire online. The same applies to the older adult OCC group, while the older adults POP group were invited to completed the questionnaire on paper. We added information on this in the section 2.1.1. AYAs – POP and 2.1.2. AYAs -  OCC, to make this more clear. 

Results, first paragraph: change "at the time of participating in the study" to "at the time of participation in the study"
Response: Thanks you. We changed this in the first paragraph of the results.

Results, table 1: Wasn't significantly significant, but there was a female preponderance, and idea why?
Response: In the Netherlands, between 1999-2016, the cancer incidence among AYAs was about 9% higher in females than in males. Our study includes a large proportion of breast cancer survivors. In addition, research has shown that  females have increased reported use of the internet for both general and health-specific purposes, which possibly also partly explains the female preponderance in our study [1]. 

Results, table 1:  Why do you think the AYA patients were off therapy a significantly longer period of time?
Response: Because of the design of the studies. The AYA POP group consists of AYAs who were already participating in the population-based SURVAYA study and diagnosed between 1999-2015 (5 to 20 years after diagnosis) and participated in the study in 2019. The older adults POP group were patients who were diagnosed between 2014 and 2016 and participated in the study in 2016/2017.

Results, table 2: There was a trend toward more internet searches for clinical trials in older adults.  Given the under-representation of AYA on clinical trials, I find this interesting and may be worth mentioning in the discussion.
Response: This is indeed very interesting and worth mentioning in the discussion. We have add this in the third section of the discussion: ‘In addition, older adults searched more often for clinical trials online compared to AYAs (37% vs 30% 0.0588). These results are in line with the underrepresentation of AYAs participating in clinical trials and increased personal barriers to cancer clinical trials enrollment [2].’  

Results, section 3.4: Remove the word "often" from the first sentence.
Response. We removed the word ‘often’. 

Discussion: Paragraph 4, line 180-change the word "was" to "did"
Response: We changed the word ‘was’ to ‘did’.

Discussion: Paragraph 5, line 291-the sentence beginning with "From the results of the study....." doesn't read well as currently written.  Please re-write to improve flow of the paper.
Response: Thank you for this comment. We re-wrote this part of the discussion: Our study results show that OCC-patients are inherently searched more often daily/several time per weeks on the Internet and were higher educated. It seems reasonable to assume that they have more health literacy skills than the general population. These factors  are important to keep in mind when conducting and interpreting future research.

References:
1.    Dee, E.C., et al., General and Health-Related Internet Use Among Cancer Survivors in the United States: A 2013-2018 Cross-Sectional Analysis. J Natl Compr Canc Netw, 2020. 18(11): p. 1468-1475.
2.    Lewin, J., et al., Evaluation of Adolescents' and Young Adults' Attitudes Toward Participation in Cancer Clinical Trials. JCO Oncol Pract, 2020. 16(3): p. e280-e289.